# A Preventive Prebiotic Supplementation Improves the Sweet Taste Perception in Diet-Induced Obese Mice

**DOI:** 10.3390/nu11030549

**Published:** 2019-03-05

**Authors:** Arnaud Bernard, Déborah Ancel, Audrey M. Neyrinck, Aurélie Dastugue, Laure B. Bindels, Nathalie M. Delzenne, Philippe Besnard

**Affiliations:** 1NUTox team, UMR 1231 INSERM/AgroSup Dijon/Univ Bourgogne-Franche Comté, 21000 Dijon, France; arnaud.bernard@u-bourgogne.fr (A.B.); deborah.ancel@orange.fr (D.A.); aurelie.dastugue@agrosupdijon.fr (A.D.); 2Metabolism and Nutrition Research Group, Louvain Drug Research Institute, Université catholique de Louvain, 1200 Brussels, Belgium; audrey.neyrinck@uclouvain.be (A.M.N.); laure.bindels@uclouvain.be (L.B.B.); nathalie.delzenne@uclouvain.be (N.M.D.)

**Keywords:** Obesity, taste, eating behavior, prebiotics, microbiota

## Abstract

Orosensory perception of sweet stimulus is blunted in diet-induced obese (DIO) rodents. Although this alteration might contribute to unhealthy food choices, its origin remains to be understood. Cumulative evidence indicates that prebiotic manipulations of the gut microbiota are associated with changes in food intake by modulating hedonic and motivational drive for food reward. In the present study, we explore whether a prebiotic supplementation can also restore the taste sensation in DIO mice. The preference and licking behavior in response to various sucrose concentrations were determined using respectively two-bottle choice tests and gustometer analysis in lean and obese mice supplemented or not with 10% inulin-type fructans prebiotic (P) in a preventive manner. In DIO mice, P addition reduced the fat mass gain and energy intake, limited the gut dysbiosis and partially improved the sweet taste perception (rise both of sucrose preference and number of licks/10 s vs. non-supplemented DIO mice). No clear effect on orosensory perception of sucrose was found in the supplemented control mice. Therefore, a preventive P supplementation can partially correct the loss of sweet taste sensitivity found in DIO mice, with the efficiency of treatment being dependent from the nutritional status of mice (high fat diet vs. regular chow).

## 1. Introduction

Development of a nutritional obesity is a complex phenomenon depending on multiple causes among which food choice plays a significant role. By determining nutrient quality and acceptability, gustation is considered as an important sensory driver of the food selection and intake. However, the relationships between taste and obesity remains poorly understood. Sweet taste sensitivity is challenged in obese rodents. Rats and mice chronically subjected to an obesogenic high fat diet (HFD) become unable to detect properly low concentrations of sweet solutions during behavioral tests minimizing post-ingestive cues (e.g., neuro-endocrine regulations) [1]. A blunting of both peripheral detection and central perception to sweet stimuli might explain this relative loss of taste sensitivity. Indeed, sucrose-evoked calcium signaling is dramatically decreased in taste bud cells freshly isolated from diet-induced obese (DIO) mice [2]. Similarly, chronic HFD elicits a down-regulation of dopamine and opioid receptors [3,4] in the mesolimbic area leading to a progressive devaluation of the reward value of oral stimuli, as found with abuse drugs [5]. Such a diet-acquired sensory deficiency might explain the tendency of DIO rodents to overeat high rewarding foods [1], probably to gain the desired hedonic satisfaction [3].

This diet-induced gustatory disorder is widely corrected in rodents after bariatric surgery, leading to healthier food choices [6,7,8]. Understanding how this improvement of the sweet taste sensitivity takes place might open new insights in obesity treatment. In rats, Roux-en-Y gastric by-pass is associated with changes in gut microbiota similar to those found after a prebiotic (P) supplementation [9], known to affect the production of hormones controlling the eating behavior, such as glucagon-like peptide-1 (GLP-1) which has an anorexigenic effect. Interestingly, behavioral responses to sweet compounds (i.e., number of licks per 10 s) are reduced in GLP-1 receptor-null mice (GLP-1 R), as compared to wild-type animals [10]. Intestinal dysbiosis is also associated with a chronic low-grade metabolic inflammation by promoting intestinal permeation of lipopolysaccharides (LPS) derived from the outer membrane of Gram-negative bacteria [11,12]. These endotoxins promote the release of pro-inflammatory cytokines in various tissues by activating members of the Toll-like receptors (TRL). A set of observations suggests that this inflammatory environment might play a role in the change of taste sensitivity in DIO mice. Indeed, taste buds express the TLR4 signaling cascade and, thus, are LPS responsive [13]. Moreover, a chronic consumption of a HFD rich in saturated fatty acids elicits a pro-inflammatory gene profile in the gustatory papillae [14]. Finally, chronic endotoxemia reduces the number of taste buds in obese mice [15]. Collectively, these findings suggest an implication of the intestinal dysbiosis in the impairment of the sweet taste sensitivity observed in DIO mice that could be improved by a prebiotic supplementation.

To explore this hypothesis, the impact of a preventive prebiotic supplementation on the orosensory perception of a sweet stimulus was compared in lean and DIO mice. Specific gut bacteria, known to be involved in the regulation of the gut peptide production and/or the gut barrier function such as *Bifidobacterium* spp. and *Akkermansia muciniphila* [16,17], were analyzed in the caecal content of mice to highlight the prebiotic effect of inulin in our model.

## 2. Materials & Methods

### 2.1. Animals

This study was carried out in the strict accordance with European guidelines for the care and use of laboratory animals and protocol approved by the French National Animal Ethic Committee (CNEA n°105). Six-weeks-old C57Bl/6 male mice were purchased from Charles River Laboratories (France). Animals were individually housed in a controlled environment (constant temperature and humidity, dark period from 7 p.m. to 7 a.m.) and had free access to tap water and chow. Experiments took place after a one-week acclimatization period. To study the impact of a preventive prebiotic treatment on the orosensory perception of sucrose during a diet-induced obesity, standard laboratory chow or custom high fat diet (Table 1) were supplemented with 10% prebiotic (P) and mice were split in four groups (*n* = 8–10): lean controls fed regular chow (C), lean controls fed supplemented regular chow (C+P), diet-induced obese mice (DIO) and supplemented diet-induced obese mice (DIO+P). Mice were fed *ad libitum* for 12 weeks. Inulin-type fructans (P—Fibruline^®^, Cosucra, Pecq, Belgium) was used as prebiotic. We have chosen a 10% prebiotic enrichment since this supplementation is known to promote metabolic [18] and cognitive benefits [19].

Evolution of the body composition (i.e., fat mass) was determined by nuclear magnetic resonance relaxometry (EchoMRI—Echo Medical Systems, Houston, TX, USA).

### 2.2. Two-Bottle Choice Tests

Tests were performed for 12 h at the beginning of the dark period in individually housed mice. Animals were food restricted during the duration of the experiment [20]. This protocol provides behavioral data combining orosensory sensations (i.e., oral detection and central perception) and post-ingestive cues. Mice were subjected to a choice between a control solution (0.3% xanthan gum in water to minimize textural influences) or a 1% sucrose in control solution. At the end of the test, fluid intake was measured for each bottle and the preference (i.e., ratio between experimental solution consumption and total intake) was calculated.

### 2.3. Gustometer

Licking behavior was studied using an original octagonal shaped gustometer of which each side has a computer-controlled shutter giving random access during a short time (10 s in the present study) to a bottle filled with a specific solution. All the bottles (five in the present study) are equipped with a lickometer. This original design, which forces the animal to move to access to the drinking source, allows a simultaneous analysis of the licking behavior, which mirrors the immediate pleasure gained from the consumption of a rewarding stimulus (i.e., “liking”) and the motivation to drink (i.e., “wanting”). Concept and procedures are detailed in [21].

In brief, 20 h water-deprived mice were subjected to two training sessions before the taste-testing sessions (30 min, each). During the first training, all the doors were opened so that the mouse had free access to all the bottles filled with water. It was a time of habituation to a new environment. During training two, the mouse learned to drink according to the protocol used during the brief-access taste testing (i.e., random opening of shutters), all the bottles being filled with the control solution (i.e., water). Each mouse had access to a first bottle for 10 s after the first lick. After this trial, all doors remained closed for 10 s before another one was opened among the 4 remaining shutters, in a randomized manner. The program continued until the animal had licked all five bottles. This event constituted a block. At the end of one block, another block started, so that the number of blocks mirrored the motivation for the stimulus. A taste-testing session was performed in water and food deprived mice to explore their licking responsiveness to a set of sweet stimuli (0.01, 0.2, 0.6, 1.0 M sucrose).

### 2.4. Blood Analysis

Freshly drawn blood samples from fasted animals were centrifuged at 6000 g for 15 min (4 °C). Plasma was collected and kept at −80 °C. Glucose, total cholesterol and triglycerides were assayed in plasma samples using commercial kits certified for in vitro diagnosis (colorimetric assays, ref#981780, 981786 and 981813) on Indiko device from Thermo (Waltham, MA, USA).

### 2.5. Gut Microbiota Analysis

Genomic DNA was extracted from the caecal content using a silica membrane-based purification technique with QIAamp DNA Stool Mini Kits (Qiagen, Hilden, Germany) according to the manufacturer’s instructions, including a bead-beating step. Total bacteria, *Bifidobacterium* spp. and *Akkermansia* spp. were analyzed by quantitative PCR, as previously described [22].

### 2.6. Statistics

Results are expressed as Means ± SEM. The significance of differences between groups was evaluated with R software (v3.4.4; The R Foundation, Vienna, Austria). We first checked that the data for each group were normally distributed and that variances were equal. We then carried out either a Student’s *t*-test or a Two-way ANOVA with the Tukey HSD post-hoc test. A principal component analysis (PCA), normalized and centered, was done with R software and the R-commander package (v2.4.4) on the different parameters studied.

## 3. Results

### 3.1. Prebiotic Supplementation Attenuates the Negative Effects Elicited by a Diet-Induced Obesity

Four groups of mice, subjected for 12 weeks to distinct diets (regulatory chow or obesogenic diet alone or supplemented with 10% P), were used to explore putative changes in orosensory perception of sweet stimuli (Figure 1A).

To verify the efficiency of this experimental design, body mass and composition, energy intake, various blood parameters, cecal tissue mass, mass of cecal content and cecal bacteria were analyzed. As expected, mice fed with the HFD displayed a greater gain in body weight and fat mass than animals fed the regular chow (Figure 1B). Prebiotic addition to the HFD led to a lower body mass gain as compared to DIO mice (Figure 1B) mainly attributable to a diminution in the relative fat mass (Figure 1C). Such a phenomenon was not observed in lean mice (Figure 1B,C). Prebiotic supplementation elicited a slight decrease in the energy intake whatever the diet, but this effect was insignificant (Figure 1D). According to previous data [23], blood glucose, plasma triglycerides and cholesterol levels were increased in DIO mice (Figure 1E–G). Surprisingly in our hands, these systemic changes were not improved in prebiotic-treated mice.

In agreement with the literature [23], chronic prebiotic consumption increased the caecal tissue mass and, in a lower extent, the fecal mass in caecum (Figure 2A,B). Prebiotic supplementation is known to modify gut microbiota composition [24]. To explore whether the bacterial signature was modified by our treatment, cecal bacterial content was studied by qPCR. As shown in Figure 2C, prebiotic supplementation tended to increase the cecal content of total bacteria whatever the diet. When abundance of selected bacterial displaying beneficial health effect was measured, a rise of *Bifidobacteria* and *Akkermansia* abundance was found in the DIO+P group (Figure 2D,E). We have previously described that ITF feeding promotes endogenous GLP-1 production through higher expression of proglucagon in the colon [25,26]. In the present study, we showed that the higher level of its expression was found in the cecal tissue from DIO+P mice (Figure 2F). Collectively, these data demonstrate the efficiency of our prebiotic protocol on DIO mice.

### 3.2. The Lower Sucrose Preference Found in DIO Mice Was Improved in Presence of Prebiotic

The preference for a sweet stimulus was explored using a two-bottle choice test. Consistent with previous published data [27], DIO mice subjected to a long-term (12 h) two-bottle preference test showed a significant lower intake (Figure 3A) and preference for 1% sucrose solution than C or C+P groups (Figure 3B), suggesting a diet-induced modification of the orosensory perception of the sweet sensation. The prebiotic attenuated the effect of the DIO. Indeed, sucrose intake was higher in DIO-P group than DIO mice, this change improving their preference score to a level similar to C and C+P mice (Figure 3B).

### 3.3. Prebiotic Supplementation Improves the Licking Behavior in Response to Sucrose Stimulus in DIO Mice

Licking behavior was studied by using the gustometer FRM8 [21]. Training sessions failed to reveal any behavioral differences between mice suggesting that DIO and/or prebiotic supplementation did not affect animal adaptability to a new environment (training 1—Figure 4A) neither their ability to learn how the device works (training 2—Figure 4B). The fact that the number of total licks was similar between groups attests that mice did not present any oromotor or mobility defect (Figure 4C).

In control mice, licking activity (licks/10 s) showed a typical dose-response curve in response to growing concentrations of sucrose, with a maximal frequency around 60 licks/10 s (Figure 5A). According to previous data [1], the number of licks/10 s elicited by sweet solutions was dramatically reduced in DIO mice as compared to controls (Figure 5A). Consistent with this observation, total licks (i.e., immediate pleasure gained from the consumption of a rewarding stimulus or “*liking*”) and number of blocks (i.e., motivation to drink, “*wanting*”) during the taste testing session (30 min) were significantly lower in DIO mice as compared to controls (Figure 5B,C). Altogether, these data confirm the existence of substantial differences in the licking responses to a sweet stimulus between C and DIO mice. A licking rate improvement was found in the prebiotic supplemented groups (Figure 5A). In response to the higher sucrose concentration, DIO+P mice displayed a similar licking response than C group, (Figure 5A), suggesting that this prebiotic treatment is able to counteract partially the orosensory deficit found in DIO mice. Despite of this significant improvement, only an upward trend in the total number of licks differentiated the DIO+P group from the DIO group (Figure 5B), the number of blocks remaining unchanged (Figure 5C).

### 3.4. Prebiotic Supplementation Brings the DIO Group Closer to Control Group

In order to better delineate the global impact of a prebiotic supplementation in control and DIO mice, a multivariate analysis (principal component analysis—PCA) was performed from variable values at the time of behavioral phenotyping (Figure 1A). Values on the *x*-axis represent the component score for the dimension 1, and those on the *y*-axis for dimensions 2- and 3, accounting for 40.50, 16.63 and 11.32% of the inertia (total = 68.45%), respectively.

Confidence ellipse analysis highlighted that C and C+P groups were partly overlapping suggesting that prebiotic supplementation did not induce any discriminant change in lean mice according to the variables studied (Figure 6A,C). They were mostly defined on the dimensions 1 and 2 by variables related to efficient taste sensations (positive correlation with high number of licks, blocks and preference) and cecal bacteria content (Figure 6B,E).

As expected, they were clearly different from the DIO group, which was mainly characterized by variables linked to obesity (cholesterol, body mass, fat mass, energy intake, glycaemia and triglycerides—Figure 6E) on the dimension 1. The third group representing the DIO+P group was found in an intermediate position between lean and obese mice suggesting that the prebiotic supplementation improved the health status of DIO mice (Figure 6A,C). Interestingly, on the dimensions 2 and 3, the DIO+P group was positively correlated with variables linked to the high caecal bacteria content and related to taste sensations (Figure 6D,E). In brief: DIO+P mice exhibited a healthier pattern than DIO mice and were closer to C & C+P mice.

## 4. Discussion

Deciphering the functional links between the nutritional obesity, taste sensitivity and eating behavior is a challenge that might open new corrective nutritional and/or pharmacological interventions to fight obesity. Nevertheless, how diet-induced obesity affects the taste perception is not yet fully understood.

Chronic consumption of an obesogenic diet (e.g., *saturated high fat diet*) leads to a shift in the gut microbiota composition and a progressive accumulation of body fat. These changes might promote a chronic low-grade inflammation (e.g., endotoxin release and production of pro-inflammatory cytokines) and a new endocrine balance (e.g., drop of GLP-1 and rise of leptin) decreasing the orosensory acuity (at taste bud level) and related reward response (corticomesolimbic level). Taken together these alterations might promote an over-consumption of energy-dense foods to compensate the sensory deficit.

According to our present knowledge, the following scenario has been proposed [28]. Gut dysbiosis and increased fat mass elicited by the chronic consumption of an obesogenic diet, induce a low-grade inflammation (e.g., LPS release and proflammatory cytokines) associated with an endocrine unbalance (e.g., low GLP-1 and high leptin levels). Collectively, these systemic changes might blunt both the taste acuity (taste bud level) and the taste-driven reward behavior (corticomesolimbic level), modifying food choices to reach an expected hedonic satisfaction. This new gustatory phenotype might lead to a preferential consumption of highly palatable foods, rich in sugar and fat, and thus to a growing obesity (Figure 7). Especially, the putative role of the gut microbiota on the taste efficiency remains to be established.

In the present study, we explored the impact of a preventive prebiotic supplementation on the orosensory responsiveness to a prototypical tastant (i.e., sucrose) in mice fed a standard chow or an obesogenic diet. Multivariate analysis clearly shows that the prebiotic supplementation poorly affects studied variables in lean mice, C and C+P groups being partially overlapping in contrast to what was found in obese (DIO) mice. Indeed, DIO+P group appeared as an independent cluster found in an intermediate position between lean and obese mice, suggesting that addition of prebiotic partially counteracts the deleterious effects of HFD. Interestingly, DIO+P group was found to be mainly depicted by variables related to gut microbiota and taste sensations. Addition of 10% inulin, not only corrected the negative effects of HFD on gut microbiota in DIO mice, but also improved their orosensory response to sweet stimuli. This last prebiotic-mediated change was found using two complementary behavioral approaches (two-bottle choice test and gustometer exploration). Collectively, our behavioral data reported a partial covery of the sweet taste sensitivity in DIO+P mice, as compared to controls, in relation with a better hedonic response to sweet stimuli (i.e., rise of lick number = “liking”) without motivational change (i.e., same block number = “wanting”) resulting in a correction of the preference deficit observed in DIO mice. A distinct impact of P supplementation on “liking” and “wanting” component of the food reward was also reported using an operant-responding performance test in DIO mice [29]. Altogether, our data raise the question of the implied mechanisms.

GLP-1 is produce in the gut by the enteroendocrine L Cells via a differential post-transcriptional processing of the proglucagon gene. This peptide hormone regulates the digestive tract (ileal brake), glucose homeostasis (incretin action) and appetite/satiation (anorexigen effect) [30]. It is known that P increase the number of endocrine L cells in the jejunum and in the colon of rodents and promote the production and release of the active forms of GLP-1, thereby decreasing glycaemia [31]. Several lines of evidence support an implication of GLP-1 in the improvement of sweet taste sensitivity after microbiota manipulation. Firstly, GLP-1 R is found both in afferent fibers innerving the gustatory papillae [32] and mesolimbic areas involved in the reward pathway (i.e., ventral tegmental area—VTA and nucleus accumbens—NAc) [33]. Secondly, GLP-1 R-null mice display a reduced response to sweet tastants during behavioral tests, suggesting that the GLP-1 signaling enhances the sweet taste sensitivity in this species [32]. Thirdly, GLP-1 reduces hedonic value of food by suppressing VTA and NAc dopamine signaling [34]. Finally, a rise of the proglucagon mRNA levels was specifically found in the caecum from DIO+P mice. Therefore, an involvement of prebiotic-induced GLP-1 release in regulation of sweet taste sensitivity is likely. Further studies combining GLP-1 R-null mice and prebiotic manipulations are required to fully establish the causal role of GLP-1 in the prebiotic impact on the gustation.

Obesity-induced endotoxemia is an alternative, but not exclusive, possible cause of implication of gut microbiota in the taste dysfunction since gustatory papillae are LPS-sensitive and display a pro-inflammatory gene profile in DIO mice [14]. However, a chronic infusion of LPS at a level similar to that observed in DIO mice was not sufficient to alter the spontaneous preference for oily solution in lean mice subjected to two-bottle choice tests. Although, the response to a sweet tastant was not analyzed in this study, it is likely that a LPS-induced low-grade endotoxemia alone does not explain the change in the orosensory perception observed in DIO mice [14].

In conclusion, this study brings the first demonstration that a gut microbiota manipulation can affect the orosensory perception of sweet compounds. These results are consistent with a recent study revealing that human volunteers shifting their habits towards a diet based on the daily consumption of P-rich vegetables showed a reduced desire to eat sweet, fatty, and salty foods, together with an increased hedonic attitude for some inulin-rich vegetables [30]. The present study also helps to reveal the complexity of the homeostatic system responsible for the proper functioning of the taste system whose disruption, undoubtedly, contributes to the establishment of nutritional obesity.

## Figures and Tables

**Figure 1 nutrients-11-00549-f001:**
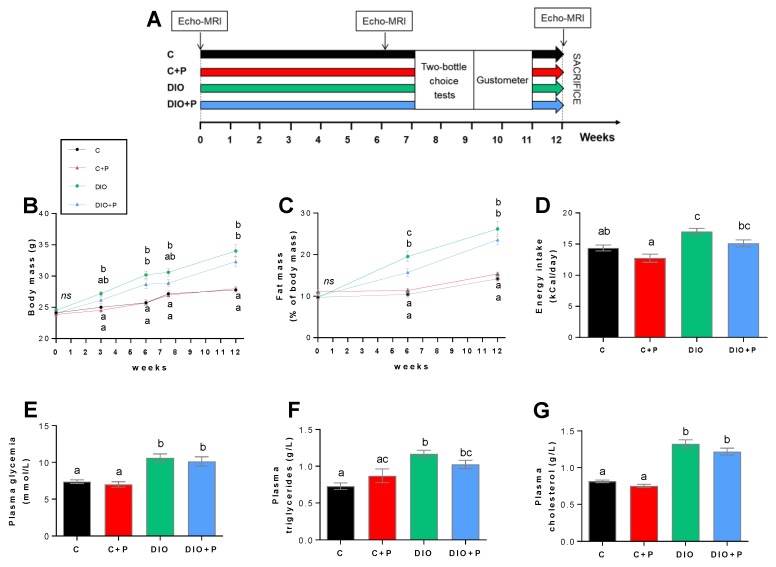
Comparison of body and biochemical parameters in mice subjected for 12 weeks to a regulatory chow or an obesogenic diet alone (C and DIO) or supplemented with 10% Prebiotic (C+P and DIO+P). (**A**) Time course of the experiment; (**B**,**C**) Evolution of the body and fat; (**D**) Daily energy intake; (**E**) Blood glucose; (**F**,**G**) Plasma triglyceride and cholesterol levels. Mean ± SEM, different letters indicate a statistical difference between groups. Significance was achieved at *p* < 0.05. C, C+P, DIO, *n* = 10. DIO+P, *n* = 8.

**Figure 2 nutrients-11-00549-f002:**
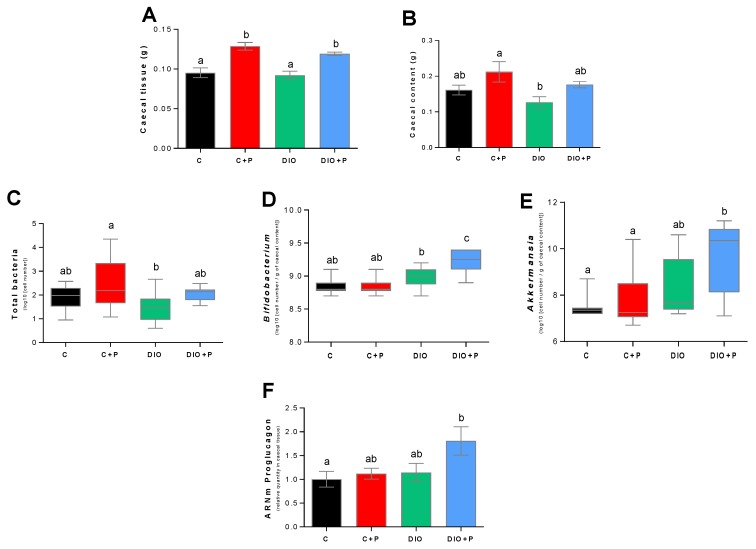
Comparison of bacterial parameters in mice subjected for 12 weeks to a regulatory chow or an obesogenic diet alone (C and DIO) or supplemented with 10% Prebiotic (C+P and DIO+P). (**A**) Cecal tissue mass; (**B**) Fecal mass in caecum; (**C**) total cecal bacteria; (**D**) *Bifidobacterium*; (**E**) *Akkermansia muciniphilla*; (**F**) relative proglucagon mRNA levels in caecum in reference to a housekeeper gene (RPL19). Mean ± SEM, different letters indicate a statistical difference between groups. Significance was achieved at *p* < 0.05. C, C+P, DIO, *n* = 10. DIO+P, *n* = 8.

**Figure 3 nutrients-11-00549-f003:**
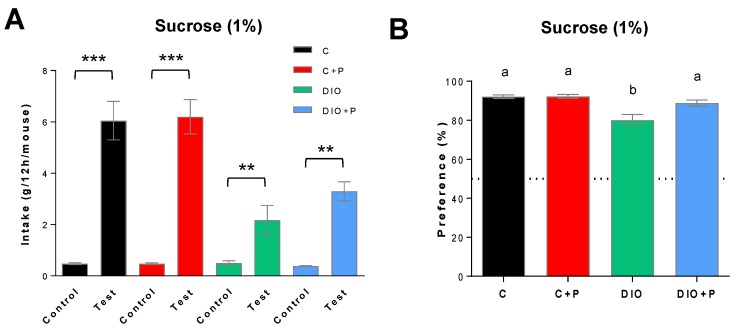
Two-bottle choice test analysis of orosensory perception of a sweet stimulus in mice subjected to a regulatory chow or an obesogenic diet alone (C and DIO) or supplemented with 10% Prebiotic (C+P and DIO+P). Animals were simultaneously subjected for 12 h to a control solution (0.3% xanthan gum in water, *w*/*v*) and a test solution containing 1% sucrose (*w*/*v*) in the control solution. (**A**) Final consumption of control and experimental solution; (**B**) Preference i.e., ratio of the final consumption of control or experimental solution upon the final total liquid intake. Mean ± SEM. (**A**) Student *t* test: ** *P* < 0.01, *** *P* < 0.001; (**B**) 2-way ANOVA + Tukey HSD: different letters indicate a statistical difference between groups. Significance was achieved at *p* < 0.05. C, C+P, DIO, *n* = 10. DIO+P, *n* = 8. The dotted line represents the 50% preference.

**Figure 4 nutrients-11-00549-f004:**
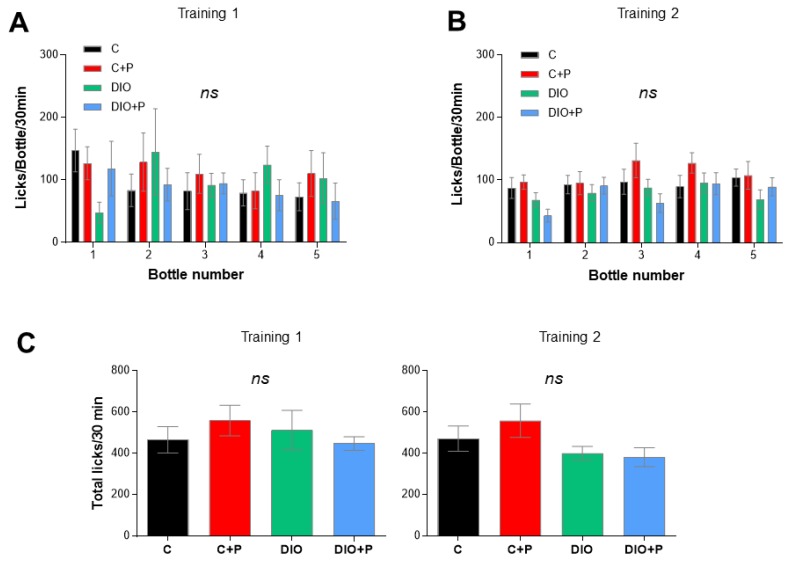
Gustometer analysis: training sessions in mice subjected to a regulatory chow or an obesogenic diet alone (C and DIO) or supplemented with 10% Prebiotic (C+P and DIO+P). 20 h water-deprived mice were subjected to 2 training sessions before the taste-testing sessions (30 min, each). (**A**) Training 1: Each mouse had a free access to the 5 bottles filled with water in order to determine the licking rate/bottle/30 min. It is time of adaptation to a new environment; (**B**) Training 2: Each learned to drink water according to the protocol used during the brief-access taste testing, i.e., a random and intermittent opening of shutters. The licking rate/bottle/30 min was determined; (**C**) Total licks for 30 min during the training 1 and 2. Mean ± SEM. 2-way ANOVA + Tukey HSD. ns non-significant. C, C+P, DIO, *n* = 10. DIO+P, *n* = 8.

**Figure 5 nutrients-11-00549-f005:**
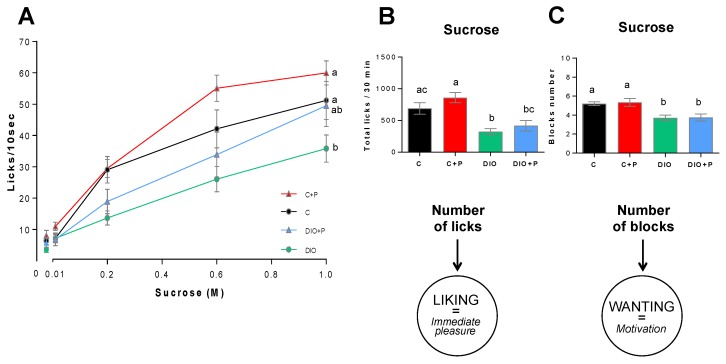
Gustometer analysis of orosensory perception in response to various concentration of sucrose in mice subjected to a regulatory chow or an obesogenic diet alone (C and DIO) or supplemented with 10% Prebiotic (C+P and DIO+P). (**A**) Brief-access taste testing responses (licks/10 s) of naïve mice to control solution (0.3% xanthan gum in water) and ascending concentrations of sucrose (0.01, 0.3, 0.6, 1.0 M). Random access to bottles was computer controlled. Zero on the x-axis represents the licking rate obtained in response to the control solution; (**B**,**C**) Total number of licks (representative of the “Liking” component) and number of blocks (representative of the “Wanting” component) performed for 30 min. Mean ± SEM. 2-way ANOVA + Tukey HSD, significance was achieved at *p* < 0.05. Different letters indicate a statistical difference between groups, ns non-significant, C, C+P, DIO, *n* = 10. DIO+P, *n* = 8.

**Figure 6 nutrients-11-00549-f006:**
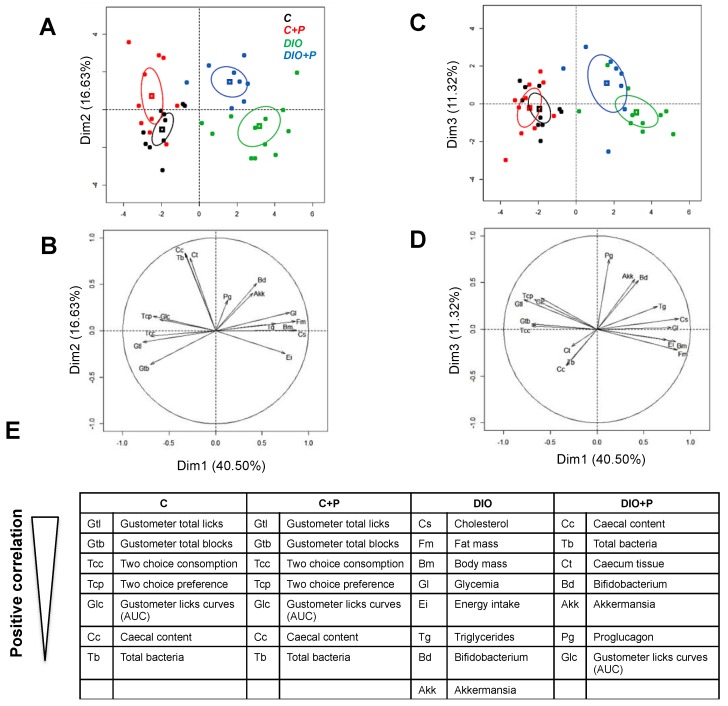
Principal component analysis performed from studied variables in mice subjected to a regulatory chow or an obesogenic diet alone (C and DIO) or supplemented with 10% Prebiotic (C+P and DIO+P). (**A**) Confidence ellipse analysis. Cluster distribution along the dimension 1 & 2. Each dot represents a mouse; (**B**) Arrows represent the direction of each variable in the 2-dimensional PCA space; (**C**) Cluster distribution along the dimension 1 & 3. Each dot represents a mouse; (**D**) Arrows represent the direction of each variable in the 2-dimensional PCA space; (**E**) Variables significantly representative of the 4 clusters (ranking in descending order of importance) and their respective abbreviations.

**Figure 7 nutrients-11-00549-f007:**
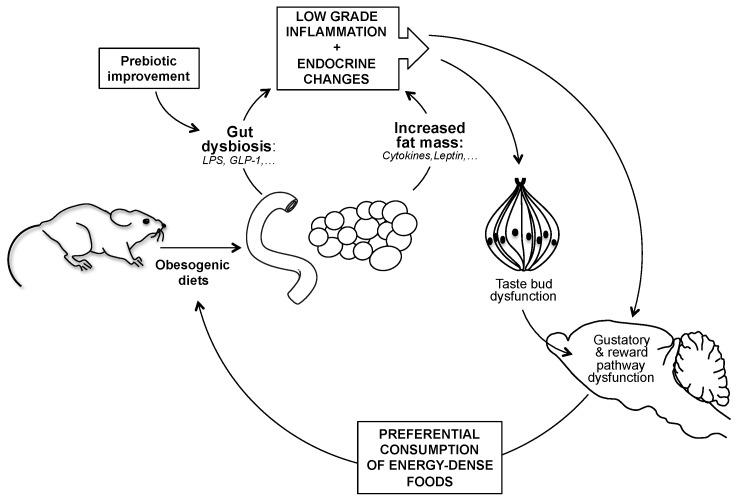
Functional relationships between diet-induced obesity and taste sensitivity in the mouse: Working model.

**Table 1 nutrients-11-00549-t001:** Composition of the diets.

Contents (% *w*/*w*)	Control Diet *(4RF21 Mucedola)*	Control Diet + Prebiotic	High Fat Diet *(4RF25 Mucedola + palm oil)*	High Fat Diet + Prebiotic
**Proteins**	18,5	16,65	15	13,5
**Carbohydrates**				
Starch	53,5	48,15	34,4	30,96
**Lipids**				
Soybean oil	3,0	2,7	2,4	2,16
Palm oil	0,0	0,0	31,8	28,62
Saturated fatty acids	0,5	0,45	16,7	15,03
Mono-unsaturated fatty acids	0,5	0,45	13,0	11,7
Poly-unsaturated fatty acids	1,3	1,17	4,5	4,05
**Prebiotic** Inulin-type fructan	0,0	10	0,0	10
**Energy** (Kcal/100 g)	315,0	315,2	505,8	506,0

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
