# Peer review of "A Preventive Prebiotic Supplementation Improves the Sweet Taste Perception in Diet-Induced Obese Mice"

_nutrients, 2019, doi:10.3390/nu11030549_

Reviewer 1 Report

The authors have conducted a study to evaluate the hypothesis that supplementation of a high-fat diet with inulin-type fructans (ITF) would prevent the loss of sweet taste perception commonly reported in diet-induced obesity in mice by altering the intestinal microbiome. 

They report that supplementation of the diet prevents in part the loss of sweet taste sensitivity observed in DIO-mice. The study is performed well, however, I had trouble following the presentation of the results, and extensive editing of English language is required. 

-it would be useful in the Introduction to provide rationale for study of bifidobacterium and Akkermansia spp only in the gut microbiota analysis

-why were only 8 mice included in the DIO-P group, whereas 10 were in the other 3 groups?

-Figure legends- the legend for the figures is unclear to me when you say "same letters indicate an absence of differences between groups". What does a, b, c etc. refer to exactly? Would be more helpful to use the letters to indicate where differences do exist

-the results are a little hard to follow and the results section should include the P-values of the comparisons. It may help to remove reference to other studies and simply describe clearly the results of this study. Comparisons can then be moved to the discussion section

-line 174 needs to be reworded, the "defect was improved by prebiotic supplementation" it seems more likely that the prebiotic attenuated the effect of HF diet

-line 203 the heading is incorrect and should be on the end of the figure legend

Author Response

Reviewer 1

1. IntroductionIt would be useful in the Introduction to provide rationale for study of bifidobacterium and Akkermansia spp only in the gut microbiota analysis

We have added a paragraph (with references) at the end of the introduction to provide rationale for study of both bacteria as follows: “Specific gut bacteria, known to be involved in the regulation of the gut peptide production and/or the gut barrier function such as Bifidobacterium spp. and Akkermansia muciniphila, were analysed in the caecal content of mice to highlight the prebiotic effect of inulin in our model.”

2. Materials & Methods

-why were only 8 mice included in the DIO-P group, whereas 10 were in the other 3 groups?

One DIO-P presented signs of illness and was sacrificed a few days after the beginning of the experiment, another one DIO-P was totally unresponsive to the diet and was a statistical outlier, we decided to remove this mouse.

-Figure legends- the legend for the figures is unclear to me when you say "same letters indicate an absence of differences between groups". What does a, b, c etc. refer to exactly? Would be more helpful to use the letters to indicate where differences do exist

By convention, multiple comparisons between more than 2 groups are always labelled with letters indicating a difference (i.e. only statistically different groups are displaying different letters). We have reformulated the sentence in order to clarify. The statistical difference was achieved at p<0.05, this written in all relevant legends.

-the results are a little hard to follow and the results section should include the P-values of the comparisons.

See above

It may help to remove reference to other studies and simply describe clearly the results of this study. Comparisons can then be moved to the discussion section

We have taken this suggestion in consideration, but we prefer to strengthen our results with others findings into the results section as suggested by reviewer 2.

-line 174 needs to be reworded, the "defect was improved by prebiotic supplementation" it seems more likely that the prebiotic attenuated the effect of HF diet

We agreed, the sentence was modified as suggested.

-line 203 the heading is incorrect and should be on the end of the figure legend

Corrected

Reviewer 2

Overall

A very nice paper, coherent, logical and clear with substatial evidence to back the main conclusion that probiotic treatment can help alleviate  sweet taste-desensitized mice of the obese mice. All of my point are minor textual corrections and require soem additional writing but overall a nice publication. i will reccomend publication with minor changes.

Introduction

General comment: While the introduction in well structured and the writing clear for the molecular or biological researcher not intimately versed in the field it would be difficult to follow and obtain all the required background to understand the significance of your findings. May I suggest adding a little more background information and linking the logic of the field as I will now suggest with a few points:

 1. The phrase 'post-ingestive cues', need further explanation for the non-expert. What exactly are these cues? 

2. In the sentence 'In rats, Roux-en-Y gastric by-pass is associated...' you mention changes in gut microbiota 'known to affect production of hormones...' Please extrapolate this fruther for the non-expert: how do these changes in the microbiota affect GLP-1 control of eating behaviour? My understanding of the main effect of GLP-1 is an an incretin hormone acting to potentiate insulin secretion in pancreatic beta-cells but it also has many other actions. Explainn this link and the current state of the field in a sentence of two.

3. Typo, pluralisation: 'Interstingly, behavioural responses to sweet compounds is ...' should be 'are'

4. The same as point 2 for the phrase 'Interestinly, behavioural responses...' What are these responses exactly? You could re-write this paragraphh leading into your hypothesis sentence by extending and linking the concepts together more tightly. This almost reads like a synopsis not an introduction. 

5. The hypothesis sentence of this paragraph begninning 'Collectively...' is clumsily written, re-write it. If you clarify and extend the introduction as a whole this sentence will become easier to write, I think. You want to prepare the reader for the main findings in a succinct and clear way: that probiotic treatment can help alleviate  sweet taste-desensitized mice of the obese mice.

6. The last sentence of the introduction should summarise the main finding(s) of your paper, which, to me, are encapsulated in Figs. 3, 5 and 6. Please do so.

Methods

1. The methods are well-written and explained overall. However, I take issue with several of your smaller analytical test descriptions. for 'Blood Analysis' and 'Gut microbiota analysis'. It is simply not sufficient or good practice to write that commercial kits were used and give no details. Commericial kits change over time and modifications are often made to the provided assay protocols. Please detail the analytical method used in these assys in detail.

2. Is xanthan gum 0.3% a standard replacement in this taste sensitivity test for sucrose solution? If so where is the reference indicating this and the validation of its use. I ask this becasue in the introduction you mention sensitivity tests obsese-induced rodents involves a 'minimisation of post-ingestive cues'. Is xanthan gum minimising these cues? Is it iso-caloric with the sucrose solution? Is the 0.3% solution isomolar with 1% sucrose? to my Knowledge xanthan gum is a polysaccharide so I assume it fulfills the requirements I have questioned here but please clarify. I ask these questions as there is a large body of research examining the relationship between carbohydrate (and other macronutrient intake) and hormonal signalling from the gut that has found significant effects from a) the macronutrient composition and b) the caloric ratios between different diets i.e. not the compositional differences. This field is involves the interplay between FGF21, incretin and insulin signalling pathways. See PMID 

24606899 ,  

25771038

Are good places to start. Please clarify these questions.

3. Please give more detail as to how the PCA analysi was conducted using R. PCA is a non-trivial statistical multi-variant analysis method that was developed from metabolomics. Important parameters to describe are: how were the axis scales created for the 2D PCA graphs: log transformations? I don't see (here or in the results) any statistical analysis conducted for multiple comparison test on the PCA individual components - are you imply taking the perviously ANOVA post-hoc values? Are you sure that all variable used for PCA are parameteric variables? These are not exclusive questions, I simply think describing more thoroughly you PCA would be good practice.

Results

1. Please explain ITF briefly in this first paragraph of the results. I known it is explained in the methods but it is so fundamental to your methodology that it deserves to be made clear to the reader why it is used in the diet here.

2. Why do you use different acronyms for the pre-biotic: use either ITF or P coonsistently throughout the results and figure legends.

3. You provide evidence that your prebiotic treatment induces bacteria displaying beneficial health effects as if all readers are initmnately aquainted with the knowledge and research of this field. Can I suggest a) adding some brief information in the introduction on Bifidobacteria etc and how and what selected bacteria are indicative of beneficial health effects. This will allow the reader to understand that your results are consistent with healthy bacterial growth in the gut whn the results are presented. Alos provide references for this. 

4. 'Moreover, proglucagon mRNA levels tended...' The link between beneficial prebiotic supplementation and increased proglucagon hormone levels in the caecal tissue is obvious and will not be to any no=expert in your particular field. Please cirrect this as per my suggestions for point 4 above.

5. Section 3.2: would you like to comment on why the overall consumption of 1% sucrose DIO mice is so much lower than in chow-fed animals? You mention in the background that mice subjected to chronic obestiy-inducing diets are reported to dsiplay lose of sweet taste sensitivity - is the simple reason? Is the extent of the decrease consitent with previous findings? A sentence in the text clarifying these points would be helpful.

6. There is something wrong with your section 3.4 heading, it appears to be showing a statistical analysis. 

6. Figure 4 legend, does '10% Prebiotic' refer to v/v or w/v ratio?

7. Figure 5 legend: you have different size fonts in this legend. 

8. There is no reference to Fig. 6D or 6E in the text. I think you have simply fogotten to add them as you do talk about the dimension 1 and 3 variables of the PCA

9. It would be beneficial to provide a summary statement for your PCA analysi of DIO vs DIO+P cohorts in section 3.5. This is very nice, conclusive data but I think you need to conclude with a summary of the Fig 6 findings, which I would say are a key component of the results. 

Discussion

I have little to add as you are the experts and it is well written. I would onyl suggest that in figure 7, which is your proposal of a model for diet-induced gut dysbiosis and inflammation in obses animals that you add you research finding from this paper i.e. suggesting that prebiotic treatment can help alleviate recover taste bud and gustatory reward function. Secondly, you mention that 'This sequence is likely incomplete...' If so please indicate on your model where the research is incomplete or inconclusive - this would make you model more visually accurate.

Author Response

Reviewer 2:

Introduction

General comment: While the introduction in well structured and the writing clear for the molecular or biological researcher not intimately versed in the field it would be difficult to follow and obtain all the required background to understand the significance of your findings. May I suggest adding a little more background information and linking the logic of the field as I will now suggest with a few points:

1.     The phrase 'post-ingestive cues', need further explanation for the non-expert. What exactly are these cues? 

We have added an exemple in the sentence.

2.     In the sentence 'In rats, Roux-en-Y gastric by-pass is associated...' you mention changes in gut microbiota 'known to affect production of hormones...' Please extrapolate this fruther for the non-expert: how do these changes in the microbiota affect GLP-1 control of eating behaviour? My understanding of the main effect of GLP-1 is an an incretin hormone acting to potentiate insulin secretion in pancreatic beta-cells but it also has many other actions. Explainn this link and the current state of the field in a sentence of two.

GLP-1 as many effects, among those it exerts an anorexigenic effect by acting on key regulatory brain areas (e.g. nucleus solitary tract, hypothalamus) and controls gastric emptying through the ileal brake (for review: Neurosci Biobehav Rev. 2017 Sep;80:457-475. doi: 10.1016/j.neubiorev.2017.06.013. Epub 2017 Jun 29.) .

A precision was added in the text.

3. Typo, pluralisation: 'Interstingly, behavioural responses to sweet compounds is ...' should be 'are'

Corrected

4.     The same as point 2 for the phrase 'Interestinly, behavioural responses...' What are these responses exactly? You could re-write this paragraphh leading into your hypothesis sentence by extending and linking the concepts together more tightly. This almost reads like a synopsis not an introduction. 

We modified the sentence according to the reviewer recommendation.

5.     The hypothesis sentence of this paragraph begninning 'Collectively...' is clumsily written, re-write it. If you clarify and extend the introduction as a whole this sentence will become easier to write, I think. You want to prepare the reader for the main findings in a succinct and clear way: that probiotic treatment can help alleviate  sweet taste-desensitized mice of the obese mice.

We modified the sentence according to the reviewer recommendation.

6. The last sentence of the introduction should summarise the main finding(s) of your paper, which, to me, are encapsulated in Figs. 3, 5 and 6. Please do so.

We don’t agree with this comment: the abstract seems to be the right place for summarization.

Methods

1.     The methods are well-written and explained overall. However, I take issue with several of your smaller analytical test descriptions. for 'Blood Analysis' and 'Gut microbiota analysis'. It is simply not sufficient or good practice to write that commercial kits were used and give no details. Commericial kits change over time and modifications are often made to the provided assay protocols. Please detail the analytical method used in these assys in detail.

We have added minor information since details about the methodology are provided by the manufacturers and can be easily consulted.

2. Is xanthan gum 0.3% a standard replacement in this taste sensitivity test for sucrose solution? If so where is the reference indicating this and the validation of its use. I ask this becasue in the introduction you mention sensitivity tests obsese-induced rodents involves a 'minimisation of post-ingestive cues'. Is xanthan gum minimising these cues? Is it iso-caloric with the sucrose solution? Is the 0.3% solution isomolar with 1% sucrose? to my Knowledge xanthan gum is a polysaccharide so I assume it fulfills the requirements I have questioned here but please clarify. I ask these questions as there is a large body of research examining the relationship between carbohydrate (and other macronutrient intake) and hormonal signalling from the gut that has found significant effects from a) the macronutrient composition and b) the caloric ratios between different diets i.e. not the compositional differences. This field is involves the interplay between FGF21, incretin and insulin signalling pathways. See PMID 24606899 ,  25771038. Are good places to start. Please clarify these questions.

The xanthan gum (XG) was used to avoid the putative textural effect of solutions containing high sucrose concentrations, a phenomenon which might interfere with the sweet taste detection and thus with liking rate.  XG was added in the same quantity in all solutions (now written in the text). XG did affect  the caloric charge of solution and as no significant impact on post-ingestives cues.

3.     Please give more detail as to how the PCA analysi was conducted using R. PCA is a non-trivial statistical multi-variant analysis method that was developed from metabolomics. Important parameters to describe are: how were the axis scales created for the 2D PCA graphs: log transformations? I don't see (here or in the results) any statistical analysis conducted for multiple comparison test on the PCA individual components - are you imply taking the perviously ANOVA post-hoc values? Are you sure that all variable used for PCA are parameteric variables? These are not exclusive questions, I simply think describing more thoroughly you PCA would be good practice.

The data were centered and normalized, we added this precision in the relevant section.

According to the reviewer comments, we have modified the results section in ordered to clarify the analysis.

Results

1. Please explain ITF briefly in this first paragraph of the results. I known it is explained in the methods but it is so fundamental to your methodology that it deserves to be made clear to the reader why it is used in the diet here. 2. Why do you use different acronyms for the pre-biotic: use either ITF or P coonsistently throughout the results and figure legends.

We modified the text according to the reviewer suggestion.

3. You provide evidence that your prebiotic treatment induces bacteria displaying beneficial health effects as if all readers are initmnately aquainted with the knowledge and research of this field. Can I suggest a) adding some brief information in the introduction on Bifidobacteria etc and how and what selected bacteria are indicative of beneficial health effects. This will allow the reader to understand that your results are consistent with healthy bacterial growth in the gut whn the results are presented. Alos provide references for this. 4. 'Moreover, proglucagon mRNA levels tended...' The link between beneficial prebiotic supplementation and increased proglucagon hormone levels in the caecal tissue is obvious and will not be to any no=expert in your particular field. Please cirrect this as per my suggestions for point 4 above.[AN1] 

We have added a paragraph (with references) at the end of the introduction to provide rationale for study of both bacteria as follows: “Specific gut bacteria, known to be involved in the regulation of the gut peptide production and/or the gut barrier function such as Bifidobacterium spp. and Akkermansia muciniphila, were analysed in the caecal content of mice to highlight  the prebiotic effect of inulin in our model.” In the result section, we have adapted the sentence as follows: “We have previously described that ITF feeding promotes endogenous GLP-1 production through higher expression of proglucagon in the colon. In the present study, we showed that the higher level of its expression was found in the caecal tissue from DIO+P mice (Fig. 2F).

5. Section 3.2: would you like to comment on why the overall consumption of 1% sucrose DIO mice is so much lower than in chow-fed animals? You mention in the background that mice subjected to chronic obestiy-inducing diets are reported to dsiplay lose of sweet taste sensitivity - is the simple reason? Is the extent of the decrease consitent with previous findings? A sentence in the text clarifying these points would be helpful.

Our data are in accordance with the provided reference (23): DIO mice display an impairment to detect and prefer sucrose solutions. This effect is not caused by the diet composition as caloric-restricted mice under an obesogenic diet display a similar behavior than lean mice fed with a control laboratory chow (23).

 The Two-bottle preference test is not able to explain the difference in the behavior of the various mice, it only describes a long term preference (here 12h) as observed in physiological conditions; this is why we also conducted  brief-liking tests (here Licks/10s) using a gustometer. Interestingly, the FRM8 gustometer device allows to explore all determinants of licking behavior including the hedonic components (i.e. liking, wanting).

6. There is something wrong with your section 3.4 heading, it appears to be showing a statistical analysis. 

Corrected

6.     Figure 4 legend, does '10% Prebiotic' refer to v/v or w/v ratio?

This is written in Table 1 into the M&M section.

7. Figure 5 legend: you have different size fonts in this legend. 

Corrected

8.     There is no reference to Fig. 6D or 6E in the text. I think you have simply fogotten to add them as you do talk about the dimension 1 and 3 variables of the PCA

Corrected

9. It would be beneficial to provide a summary statement for your PCA analysi of DIO vs DIO+P cohorts in section 3.5. This is very nice, conclusive data but I think you need to conclude with a summary of the Fig 6 findings, which I would say are a key component of the results. 

A sentence was added according to the reviewer suggestion.

Discussion

I have little to add as you are the experts and it is well written. I would onyl suggest that in figure 7, which is your proposal of a model for diet-induced gut dysbiosis and inflammation in obses animals that you add you research finding from this paper i.e. suggesting that prebiotic treatment can help alleviate recover taste bud and gustatory reward function. Secondly, you mention that 'This sequence is likely incomplete...' If so please indicate on your model where the research is incomplete or inconclusive - this would make you model more visually accurate.

According to the reviewer suggestion, this sentence was removed in the text and the figure was modified.